# Influence of Drainage on Peat Organic Matter: Implications for Development, Stability, and Transformation

**DOI:** 10.3390/molecules25112587

**Published:** 2020-06-02

**Authors:** Lech W. Szajdak, Adam Jezierski, Kazimiera Wegner, Teresa Meysner, Marek Szczepański

**Affiliations:** 1Institute for Agricultural and Forest Environment, Polish Academy of Sciences, 60-809 Poznań, Poland; teresa.meysner@isrl.poznan.pl (T.M.); morfeush@tlen.pl (M.S.); 2Faculty of Chemistry, University of Wrocław, 50-383 Wrocław, Poland; adam.jezierski@chem.uni.wroc.pl; 3Faculty of Agriculture and Biotechnology, University of Science and Technology, 85-029 Bydgoszcz, Poland; kwegner@utp.edu.pl

**Keywords:** drained and undrained peatlands, peats, humic acids, thermal, paramagnetic and optical properties

## Abstract

The agricultural use of peatlands, the stabilization of the substrate for building or road construction or for increasing the capacity of soil to support heavy machinery for industrial activities (peat and petroleum extraction), harvesting to provide peat for energy, and the growing media and initiation of chemical processes must be preceded by drainage. As a consequence of drainage, peat underwent an irreversible conversion into moorsh (secondary transformation of the peat). The object of the study was to investigate comparatively the organic matter composition and molecular structure of humic acids (HAs) in the raised bog, fen, and peat-moorsh soils developed in various compositions of botanical cover, peat-forming species, and oxic and anoxic conditions as a result of the oscillation of ground water during drainage as well as to evaluate the vulnerability of soil organic matter (SOM) to decomposition. Drainage was shown to be the principal factor causing the various chemical compositions and physicochemical properties of HAs. Large and significant differences in chemical composition of peat and the properties of HAs were found to be related to the degree of decomposition. The HAs from drained peatlands were less chemically mature. In contrast, the HAs from fen and raised bog were found to be more mature than that of the corresponding drained peatlands. The above findings showed the distinguishable structure of HAs within the soil profile created by the plant residue biodegradation and formed in both oxic and anoxic conditions. The analytical methods of thermal analysis together with the optical densities and paramagnetic behaviour are suitable and effective tools for studying structure–property relationships characterizing the origin and formation process of HAs in various environmental conditions.

## 1. Introduction

There are peatlands in every climatic zone. Total Earth’s area of these elements of the landscape is more than 3 million km^2^ (about 3% of the Earth’s land surface). Peatlands accumulate some 70% of natural freshwater. Each year, they absorb 0.37 giga- tonnes of carbon dioxide (CO_2_), which corresponds to a greater capacity for carbon storage than that attributed to all other vegetative organisms in the world [1]. 

The conversion of vegetable matter into peat structures changes depending on the vegetation type, ability of tissues to be decompose, deepness of roots and water table, pH, ionic strength, balance between humification and degradation processes and also depends on access to organic and inorganic compounds [2,3,4,5,6,7,8,9,10].

Over 20 forms of peat bogs, 19 forms of peat bogs and about 6 forms of swamps could be treated as peatlands [11]. Ombrotrophic bogs are mainly rain fed and have nutrient deficiency. Fens supplied by surface and ground water are richer in nutrients then bogs [12,13]. Bogs and fens grow in conditions of full water saturation what significantly slows down a degradation of organic matter.

The peatlands must undergo drainage for their application in agriculture as a soil substrate and for their stabilization as substrate in building and road construction as well as a support for heavy machinery for industry used e.g. in peat and petroleum extraction for energy producing [14,15,16]. 

In wetlands located in the cool climate based on an average storage rate of 200 kg C ha^−1^ and assuming an area of about 350 × 10^6^ ha. The annual accumulation not disturbed wetlands has been calculated as 0.06–0.08 Pg C y^−1^ [17]. The total drained area in the period 1795–1980 was 8.219 × 10^6^ ha; 5.5 × 10^6^ ha; 9.4 × 10^6^ ha for crops, pastures, and forests, respectively.

Up to 35 × 10^6^ ha of wetlands has been drained [18]. However, in the following countries the rate of subsidence of drained organic soils ranged from 1 to 8 cm y^−1^ (The Netherlands: 1.75 cm y^−1^, Quebec in Canada: 2.07 cm y^−1^, Everglades in the United States: 3 cm y^−1^, San Joaquin Delta in the United States: 7.6 cm y^−1^, and Hula Valley in Israel cm y^−1^). Drying shrinkage, loss of the buoyant force of groundwater, compaction, wind erosion, burning, and microbial oxidation belong to the main reasons for subsidence of histosols.

As reported by Terry [19] approximately 73% of the loss of surface elevation in Everglades histosols is caused by microbial oxidation. In addition, an assumed C release from drained wetlands by oxidation of the organic material of 10 t C ha^−1^ y^−1^ a global annual C release is 0.05 to 0.35 Pg C. In gleysols the long-term drainage of 106 ha impacts on an extra release of 0.01 Pg C y^−1^. Total release from histosols and glaysols ranges from 0.03 to 0.37 Pg C y^−1^.

In the period 1795–1980, about 4% of the wetland was drained in the tropics. In cool regions, the annual shift (loss of sink strength and gain of source strength) in the global C balance is 0.063–0.085 Pg C due to drainage of histosols. However, including tropical histosols, the global shift would be 0.15–0.184 Pg C y^−1^ [18]. Under cultivation the potential to increase C levels in soils is largely promoted to upland soils. Restoring C sinks in wetland soils drained artificially is improbable unless they are transferred from agricultural production to natural wetlands [20].

At present time, a total amount of decomposed worldwide peatlands is equal to 65 million ha. The drainage is the key driver of the degradation of peat soils [21]. From 14 to 20% of peatlands in the world and 14% in Europe are applied for agriculture. The meadows and pastures are the great majority of peatlands used for agriculture in Europe. The percentage use of European peatlands for cultivation is as follows: Hungary (98%), Greece (90%), The Netherlands (85%), Germany (85%), and Poland (70%). 

In countries of other continents e.g. Canada the national resources of peatlands were mostly undrained and forested; only 15% were drained and used for agriculture. In USA, more than 230,000 hectares of fen in the Florida Everglades are mainly used for cultivation of sugar cane and rice. 20% of Indonesia’s peatlands were drained and utilized for agriculture. In recent years due to increasing nature protection and for economic cause the total area of agricultural peatlands has reduced [12,22].

Drainage enhances the emission of CO_2_ and N_2_O and decreases emission of methane from the peat. The evolution from 15 to 17 Mg CO_2_ from grassland and of 41.1 Mg CO_2_ from ploughed fens was determined. In addition, drained peatlands emit of nitrous oxide with fluxes varying from 2 to 56 kg N_2_O–N ha^−1^ year^−1^, while CH_4_ fluxes ranged from −4.9 to 9.1 kg ha^−1^ year^−1^ [23].

Drainage, besides the impacts on the mineralization of organic matter and the evolution of gasses reduces also the diversity of peatland vegetation and favours forest plant species. The first species to decrease and vanish are those that thrive on wet lawn and flark levels. The plants related to drier hummocks should adapt to changing conditions and initially even benefit from drainage. Later, the growth of tree stands and increased shade will limit their opportunities to thrive. More mature tree stands will also lose more water through evapotranspiration, increasing the drying-out effect, and accelerating the mineralization of organic matter [21,24]. The rate of the changes caused by drainage will depend on factors including: the concentration of mineral and organic nutrients, availability of moisture, the efficiency of the drainage ditches, and intensity of tree growth [24,25]. Typical members of the plant kingdom are resistant to drying stress [26] whereas wetter and more nutrient-rich peatlands are changed more dramatically [27,28].

Maslov et al. [29] published the basic information on eighty peatland experiments which were established at various times within the territory of the Russian Federation. The aims of these field experiments were to study the peatland ecosystem as well as site transformation under the influence of drainage, forestry, and agricultural use. Materials obtained on the experimental sites include multidiscipline investigations and characterize wetlands in general, peat soil properties, drainage system arrangements, etc.

Annual raising of the Carex peat in eastern Europe is from 0.5 to 1 mm per year, while in northern mires, the Carex peat layer had been growing slowly and thus almost two meters of peat has accumulated over 9800 years (~0.2 mm per year) [30]. The drainage of peatlands impacts on the lowering of this element of the landscape. In New Zealand and in Norway the lowering is 3.4 cm year^−1^, and 2.5 cm year^−1^, respectively [31,32]. Thus, deposits that have accumulated over many millennia can disappear over a time scale that is very relevant to human activity.

Last advances in peatland restoration methods have led to the establishment of *Sphagnum* moss on the remnant cutover peat surface following peat extraction; however, evaluating restoration success remains a key issue. The study of Lucchese et al. [33] showed an increase of organic matter accumulation from 2.3 ± 1.7 cm 4 years post-restoration to 13.6 ± 6.5 cm 8 years post-restoration. For comparison, at an adjacent non-restored section of the peatland organic matter, accumulation was significantly lower (*p* < 0.001 for all years), with mean thicknesses of 0.2 ± 0.6 and 0.8 ± 1.2 cm for 24 and 28 years post-extraction, respectively.

The changes of oxic conditions in peatlands as a result of the oscillation of ground water during melioration activates the irreversible conversion of peat into moorsh (synonyms: peat-moorsh soils, secondary transformed peat, and mucks) which is usually treated as the secondary transformation of peat [34]. This conversion is responsible for the changes in the structure of organic mass constituting these soils, causing modification of the properties of high molecular weight substances from hydrophilic to hydrophobic, the disappearance of peatland, and a decrease of anisotropy of peat deposit [35,36,37,38]. As the result of the above processes the soil’s abilities to swell again, to disperse, and hence to re-soak are lost as well. The moorsh formed from peat seems to be fine-grained, more colloidal, and degraded due to particle size and a higher percentage of mineral matter [39,40].

The irreversible loss of wettability due to drying is responsible for damage of colloidal behavior in peat. The secondary transformation of peat showed the disintegration of the thermodynamic equilibrium in peat. The decline in peat soil moisture content resulting from drainage implicates shrinkage of the peat structures. Volume change from shrinkage is generated by several forces acting at the microscale, whereby its mechanism and magnitude differ from those in mineral (clay) soils. Drying and wetting of peat soils giving soil volume changes is manifested in soil vertical movement and bulk density changes [41,42,43]. In addition, biotic and abiotic conversions and degradation of peat organic matter is observed as effect of drainage [44,45,46,47]. Säurich et al. [15] indicated that bog peat samples tend to be more sensitive to anthropogenic disturbance than fen peat samples.

Drainage is the main direct cause of fen habitat degradation, either due to reclamation of fen or the changes in the water flow within fen systems. Reducing of the water level in peatlands activates anaerobic conditions into aerobic ones and accelerates the peat mineralization. In the first period after drainage, this causes usually an increase of nutrient availability, especially nitrogen and phosphorus, which are released during mineralization [48]. However, the raised fertility is usually only a short-term effect [49], and therefore, additional fertilization is needed to sustain economically prospective agricultural production on drained fens, which has a further negative impact on biodiversity. Apart from increasing nutrient availability, drainage also lowers the water storage capacity of peat soils, making them more susceptible to water-table fluctuations and droughts. A further aspect of fen habitat degradation is acidification. This may be related to drainage, which results in partial replacement of groundwater by rainwater [50], and to the increased atmospheric deposition of nitrogen and sulphur compounds [51].

In general, drainage of peat leads to the progressive differentiation of the hydrophobic peptides and total amino acid content in organic matter. In proteins of peats, hydrophobic contacts exist between hydrophobic and hydrophilic structural elements (between the side chains of the radicals of phenylalanine, leucine, isoleucine, valine, proline, methionine, and tryptophan). Hydrophobic forces stabilize the tertiary structure of proteins and determine the properties of lipids and biological membranes. The presence of amino acids, hydrocarbon chains, and other nonpolar fragments in their composition are related to hydrophobic properties of humic substances [37,52].

Since organic matter is a major component of the soil phase of peat and moorsh soil causing soil water repellency, it is important to study the effect of chemical soil properties on their wettability. It was observed that the significant changes of chemical properties of transformed organic matter in peat have a significant influence on the sequential modifications in physical and hydraulic features created by lowering of water table for agriculture. Van Dijk [53] postulated relationship between a high increase of shrinkage, changes in the number of many chemical and physical properties and humic components in peats [53].

The retention of water by peat can be considered in terms of reaction of water molecules with the surfaces of peat particles. The phenomenon is therefore amenable to the analytical methods of colloid and surface chemistry. Considerable attention should be paid to the chemical composition of peat, identifying the molecular structure of substances and aggregates most likely to hold water strongly. Knowledge of this structure can provide a rationale for treatments intended to remove or render less water-retentive the most hydrophilic fraction.

The substances of greatest immediate interest are humic acids (HAs). HAs are created in peats by degradation, poly-condensation, polymerization, and poly-addition of organic substances as a result of habitat and anthropogenic processes, including the degradation of plants and animal residues which are characterized by a complex macromolecular structure with aromatic and aliphatic units; peptide chain; and nitrogen in aliphatic, cyclic, and aromatic forms. The HAs represent macromolecular polydisperse biphyllic systems, including both hydrophobic domains (saturated hydrocarbon chains and aromatic structural units) and hydrophilic functional groups, i.e., having an amphiphilic character. The hydrophilicity of peat surfaces is generally attributed to the availability of organic functional groups capable of hydrogen-bonding. Such groups, well known to organic chemists, include carboxyls as well as phenolic and alcoholic hydroxyls. HAs have been regarded as peat component principally responsible for water retention in peat. The hydrophilicity of HAs depend not only on the numbers of hydroxyl and other polar units but on their ability to form hydrogen-bonding with water as there are hydroxyl groups inaccessible to water.

These substances are formed of similar but not identical substrates; therefore, no two HAs are identical. The HAs from various types of peats and organic and inorganic components of HAs matrices are altered to different extents, significantly differing both in composition and properties. The quantity and quality of HAs in soils organic matter depend on the balance between primary productivity and the rate of decomposition [38]. Chemical properties and structural characteristics of humic substances were shown to be better predictors of soil organic matter turnover rate in vertisols than soil organic matter content.

This suggests the possibility of using humic substances as indicators of soil organic matter turnover because they are sources of intermediates and energy for many chemical and biochemical pathways in the soil [39,40].

The study outlined here was conducted to propose tools and analytical methods for the quantitative and qualitative evaluation of the turnover processes occurring in the oxic and anoxic conditions of developed peat deposits.

The aim of this study is to analyse comparatively the organic matter composition and molecular structure of HAs in the raised bog, fen and peat-moorsh soils developed in various composition of botanical cover, peat-forming species, and oxic and anoxic conditions as a result of the oscillation of ground water during drainage as well as to evaluate the vulnerability of soil organic matter (SOM) to decomposition.

## 2. Methods

### 2.1. Study Sites

Six sites from 3 peatlands were found to vary according to their macrofossil analysis, their state of decomposition, and their GPS parameters, and an outline of their location has been given (Figure 1 and Figure 2, Table 1).

### 2.2. Collection of Smples (WRB classification 2015)

Peat samples were collected from the following:(a)Baltic-type raised bog (Kusowo)(b)fen (Stążka)(c)peat-moorsh soils: Ch1, Ch2, Ch3, and Ch4 (Turew)

Peat samples were collected in triplicate from the field using a 5.0-cm diameter Instorf peat auger and various depths from 0 to 100 cm in the stratigraphic profile of each peat deposit, transported to the laboratory at ca. 4 °C and stored at −20 °C. The samples were dried at 20 °C and homogenized in a grinder after removal of any visible live plant material, after which they were passed through a 1-mm sieve to remove rock fragments and large organic debris. The botanical composition of peat was analyzed by the microscopic method and subsequently classified according to the Polish standards (PN-76/G-02501 1977). Peat samples were used for the description, classification, and physicochemical analysis [54] (Table 1).

The Kusowo Bog is situated in the West Pomeranian Voivodship. This is likely the best-preserved Polish Baltic-type raised bog. The reserve has an area of 326.56 ha and is entirely included into the Szczecineckie Lake Natura 2000. The mean peat thickness may exceed 12 m. The mean annual air temperature is 7.2 °C. The mean annual precipitation is 760.1 mm. (Table 2) (Figure 1). The age of peatland is 710 years (Table 3). Moisture contents of peat samples ranged from 89.69% to 92.31% (Table 5). Among raised bogs, that of the Baltic-type is distinguished, one peculiar to a humid climate, with a high rainfall. The bog, formerly a lake, developed on a moraine with kames in an extensive depression. The bog is clearly divided into two parts—northern and southern—separated by three mineral mounds.

The northern part is better preserved, with a gently sloping “living” dome, about 3–4 m high, with some peat ponds. The dome is mostly composed of *Sphagnum* hummocks and hollows of the plant association *Sphagnetummagellanici*. In places with high water content, e.g., near peat ponds, the *Rhynchosporetumalbae*, *Eriophoroangustifolii-Sphagnetumrecurvi*, and *Caricetumlimosae* are located. Nearly half of the northern part consists of wooded habitats, primarily pine and birch bog forests (*Vacciniouliginosi-Pinetum* and *Vacciniouliginosi-Betuletumpubescentis*) [55,56] (Figure 1)

The Stążka fen is a part of the “Bagna nad Stążką”—a mire complex located in Northern Poland in the region of the Tuchola Forest. The fen is part of the Nature Reserve, where the whole complex of natural peatlands is under protection. This peatland measures some 478.45 ha. The maximum thickness of peat deposits is 1.4 m. This region was characterized by a mean annual air temperature of 7.2 °C and by mean annual precipitation 598.9 mm (Table 2) [26]. The age of peatland is 1400 years (Table 4). Moisture contents of peat samples ranged from 93.15% to 93.92% (Table 5).

Peat-moorsh samples were taken from four chosen sites marked as Ch1, Ch2, Ch3, and Ch4 on the 4.5-km long transect of peatland located in the Chłapowski Agro-ecological Landscape Park of the West Polish Lowland, about 40 km southwest of Poznań. These are the sites investigated along the Wyskoć Canal. The thickness of peat deposit ranges from 1.5 to 2.75 m. The mean annual air temperature is 8.4 °C, the mean annual precipitation is 539.3 mm (Table 2), and the growing season is about 200 days (Figure 1 and Figure 2). Moisture content of peat samples ranged from 61% to 86.82% (Table 5).

All examined samples were classified as raised bog, fen, and peat-moorsh soils according to World Reference Base Soil Resources (2015) [57].

### 2.3. Organic Carbon, Nitrogen, C/N, and Degree of Decomposition

Total carbon (TC) and inorganic carbon (IC) concentrations in dried peat samples were analyzed by means of a Total Organic Carbon Analyzer (TOC 5050A) with a Solid Sample Module SSM-5000A, Shimadzu, Japan. Total Organic Carbon (TOC) was analyzed by placing about 50 mg of a soil sample in Total Organic Carbon Analyzer (TOC 5050A) with Solid Sample Module (SSM-5000A) produced by Shimadzu (Kyoto, Japan). Total organic carbon (TOC) was calculated as the difference between total and inorganic carbon.

Air-dried peat samples for the measurements of hot water extractable organic carbon (C_HWE_) were mixed with deionized water and heated at 100 °C for two hours under a reflux condenser. Extracts were filtered through 0.45-μm pore-size filters and analyzed on TOC 5050A facilities, Shimadzu, Japan [58].

Total nitrogen was evaluated by the Kjeldahl methods, using Vapodest 10s analyser (Gerhardt, Germany). N-total was evaluated by semimicro-Kjeldahl method; 0.5 g of soil sample was set in Kjeldahl digestion tube. Five mL of twice distilled water was added. After 30 min, 2.5 mL of 95% H_2_SO_4_ and powder zinc were added and heated 20 min. Fifteen mL of 95% H_2_SO_4_, 5 g of K_2_SO_4_, and a small amount of selenium mixture were added and heated at 350 °C. A Kjeldahl digestion tube was cooled to room temperature. Mineralized sample was set in a 100-mL volumetric flask and filled to the line by twice distilled water. A mineralized sample was centrifuged (MLW K 23 D, Germany) in 3000 rpm by 15 min. Twenty-five mL of the sample was set in Parnas–Wagner apparatus. Fifty mL of 30% NaOH was added. Fifty mL of distillate was collected to a 100-mL Erlenmeyer flask filled by 20 mL of 4% H_3_BO_3_ and indicator (bromocresol green and methyl red mixture). The blue distillate was titrated to red color by 0.02 N HCl.

The atomic ratios of C/N were calculated after determining the concentrations of carbon and nitrogen.

Peat decomposition was determined at the sampled depths according to the field squeezing method by means of the von Post classification scale [59]. This method identifies ten classes of decomposition, with H1 being undecomposed peat and H10 being completely decomposed peat. Peat type was determined based on plant macrofossil analysis [60].

### 2.4. Extraction and Purification of HAs

Isolation of HAs was performed according to the recommendations of the International Humic Substances Society procedure under relatively gentle conditions [61,62]. The humic substances were extracted from air-dried peat soils using the grain size fraction smaller than one millimeter with 0.1 M NaOH using an extractant/peat ratio of *v*/*v* 5:1) at pH 7.00 under a N_2_ gas atmosphere and shaken for four hours and, after that, were stored overnight for the coagulation of HA fractions which were separated by centrifugation. The purification of the HAs was performed by the following method. The suspension was centrifuged at 4000× *g* at 24 °C for 1 h. The solution was acidified with 6 *M* HCl to pH 1.3 to precipitate the HAs and was allowed to stand overnight. Then, the solution was centrifuged to eliminate the supernatant. The procedure was repeated three times. Finally, precipitated HAs were freeze dried and stored in a vacuum desiccator over P_2_O_5._

Moisture analyzer MAX series (Radwag, Poland) was used to determine hygroscopic humidity content in peats. The content of hygroscopic humidity was taken into account in the analysis of peats.

^14^C dates were provided by the Poznan Radiocarbon Laboratory (Poland).

### 2.5. Elemental Composition

The content of C, N, H, and S in every HAs fraction was analysed using the Vario Micro Cube Elemental Analyser. The samples were burned in an oxygen atmosphere at 1150 °C. The gases (CO_2_, N_2_, H_2_O, and SO_2_) were separated chromatographically and measured by a thermal conductivity detector (TCD). Oxygen content was obtained by subtracting the sum of other elemental contents from 100%, and the results were expressed as percentages calculated on the basis of total element amounts (Table 6).

### 2.6. VIS-Spectroscopy of HAs

E_4_/E_6_ ratios were determined by dissolving 3 mg of HAs in 10 mL of 0.05 M NaHCO_3_ (pH = 9.0) and by measuring optical densities at λ = 465 nm (E_4_) and λ = 665 nm (E_6_) spectrophotometrically on SHIMADZU UVmini-1240 (Japan) with 1 cm thickness (Table 7) [63].

### 2.7. Electron Paramagnetic Resonance of HAs

Electron paramagnetic resonance (EPR) X-band (9.8 GHz) was recorded at room temperature on a Bruker ElexSys E500 instrument equipped with an NMR teslameter ER 036TM and E 41 FC frequency counter. The concentration of spins was measured using a double integral procedure applying the Bruker WinEPR program. Leonardite was used as a spin concentration standard (International Humic Substance Society) [64].

### 2.8. Differential Thermal Analysis of HAs

Thermal properties of HAs were evaluated by means of an OD-103 derivatograph (MOM-Paulik-Paulik-Erdey, Hungary) [65]. The curves of differential thermal analysis (DTA), thermogravimetry (TGA), and differential thermogravimetry (DTG) were recorded simultaneously. Weight losses at different steps of thermal decomposition were calculated from the TGA curves. The Z index was calculated, and aliphatic character of HAs was pointed out. All chemical analyses were run in triplicate, and the results were averaged.

### 2.9. Statistical Analysis

The confidence intervals were calculated using the following formula: $\bar{x}$ ± t_α (n − 1)_ SE, where: $\bar{x}$ is the mean; t_α (n − 1)_ is the value of the Student test for α = 0.05; n−1 is the degree of freedom; and SE is standard error. Linear correlations between the values were calculated. Normal distribution of the results and homogenous variances were checked before statistical analysis.

Principal component analysis (PCA) using Statistica version 9.1 was performed to determine the correlations between Baltic-type raised bog, fen, and peat-moorsh soils in the peat deposit properties and physicochemical properties of soil organic matter. The number of factors extracted from the variables was determined by a scree test according to Kaiser’s rule. With this criterion, the first two principal components with an eigenvalue greater than a third were retained. Principal component analysis (PCA) using Statistica version 9.1 was performed to determine the correlations between raised bog, fen, and peat-moorsh soils and of peat deposits properties and physicochemical properties of soil organic matter.

## 3. Results and Discussion

### 3.1. Characteristics of Peats

#### 3.1.1. Decomposition Degree, Carbon, Nitrogen, and C/N Ratio of Peat Soils

##### Decomposition Degree

Degree of decomposition is an important property of the organic matter in soils and other deposits which contain fossil carbon. It describes the intensity of transformation or the humification degree of the original living organic matter. Our knowledge of the degree of decomposition and chemical characteristics of peat soil may be translated into peat physicochemical properties, therefore, both the degree of decomposition and type of peats described above [7,66].

All peats were decomposed, however, in varying degrees (Table 1). A gradual increase of the degree of decomposition has been observed in the order from the Baltic-type raised bog (H2–H5) throughout fen (H3–H5) to peat-moorsh soils (H7–H8). A drop of the water level in the peat-moorsh soils increased the oxygen content and accelerated the mineralization of organic matter, leading to an increase of the degree of decomposition (H8). Therefore, a decrease in the degree of peat decomposition is treated as an indicator of a rise of the ground water table [67].

During decomposition of organic matter, labile compounds such as peptides and polysaccharides are preferentially decomposed while refractory aromatic or aliphatic compounds become residually enriched [68].

Almost all nitrogen in organic matter is in organic form (0–90%). Nevertheless, the chemical composition of nitrogen in organic soil fraction is not completely understood and little is known of the factors affecting the distribution of organic nitrogen forms in soils.

Humus is composed from 20% to 60% of HAs. The nitrogen (20–40%) in HAs consists of amino acids or peptides, the main unit of proteins, and is connected to the central core by hydrogen bonds [69]. Therefore, amino acids and proteins are the main fraction of total nitrogen in organic soils.

Soil amino acids are components of protein conglomerates. They occur in stable form. Soil proteins included in organic colloids are hydrophilic colloids. These colloids are water-related. Drainage causes the denaturation of colloids and a change in the properties of proteins from hydrophilic to hydrophobic. During the drainage process, a progressive increase in the hydrophobic amino acid content was observed. In proteins of peats, hydrophobic contacts exist between hydrophobic and hydrophilic structural elements (between the side chains of the radicals of phenylalanine, leucine, isoleucine, valine, proline, methionine, and tryptophan). The presence of amino acids, hydrocarbon chains, and other nonpolar fragments in their composition are related to hydrophobic properties of humic substances [37,52].

##### TOC

Significantly increasing concentrations of TOC exhibited a trend with a gradually increasing down-core for all investigated sites (Table 1). A lower TOC was determined in peat-moorsh soils in comparison with Baltic-type raised bog and fen. This relationship was found to be related to the degree of decomposition. These changes observed for the degree of decomposition, and TOC could have been caused by a change in any one of a number of the following factors, e.g., water level and the balance between accumulation/decomposition, indicating most likely that they are attributable to the combination of such factors.

The lower TOC content in the upper layer may be related to the increased rate of decomposition rather than accumulation. In deeper layers of peat profile under saturated conditions, lower decomposition rates of organic matter are observed.

Our results are in line with the results of Benavides [70], who showed that peat and carbon accumulation rates were lower in drained sites, indicating either greater decomposition rates of the upper peat column or lower production by the changed plant communities. The ecological services offered by peatlands to agrarian communities downstream are important. In addition, he observed that species composition was much affected by drainage, which resulted in a reduction in cover of *Sphagnum* and other peat-forming species, and by the encroachment of sedges and Juncus effusus. The ability of peat to store water and carbon was also reduced in drained peatlands. Vegetation records show a shift towards sedge-Juncus communities around 50 years ago when agricultural use of water increased.

Batjes [71], Chambers et al. [72], Fontaine et al. [73], and Wang et al. [74] suggested that a lack of a supply of fresh carbon may prevent the decomposition of organic carbon pool in deeper layers of peat deposits.

##### C_HWE_

C_HWE_ is responsible for the microbiological activity in depth profile to be exuded from plant roots, acting as a significant source of carbon fueling microbial metabolism [75]. This fraction is a potential source of carbon and energy for heterotrophic organisms and contributes significantly to stream ecosystem metabolism. The flux of C_HWE_ from an ecosystem can be a significant component of carbon (C) budgets, especially in watersheds containing wetlands. In watersheds containing organic wetland soils or peatlands, the flux from the watershed can be 4–8% of annual net primary production, a significant fraction that should be addressed when performing a carbon mass balance [76].

A significant decline of C_HWE_ with increasing depth in all sites was observed (Table 1). The highest decrease of C_HWE_ (68.2%) with increasing depth was measured in Ch2, representing peat-moorsh soil, and can be attributable to the most degraded peat-moorsh soil.

In contrast, the lowest decrease of C_HWE_ (25.4%) with increasing depth was determined in the lowest decomposed peats of Baltic-type raised bog as compared to that of peat-moorsh soils. Thus, C_HWE_ can be said to increase gradually with the increase of peat during decomposition (Table 1).

The decreases in C_HWE_ concentrations were generally accompanied by the increases of the TOC in all sites of sampling. We found a consistent pattern in parallel changes as per decreasing/increasing C_HWE_ and TOC concentrations among all sites.

The values of the ratios C_HWE:_/TOC in undrained peat (raised bog and fen) ranged from 0.9 to 2.7, while in drained peat, it ranged from 1.2 to 4.1. The highest value of the ratio, 4.1, was measured in the Ch2, which showed the highest value of the degree of decomposition.

Monitoring the properties of C_HWE_ in the peat profile is frequently used to evaluate changes in peat quality and to explain shifts in peatland ecosystem function [77].

The purification of ground water by the transect consisting of peat-moorsh soils of 4.5 km length was observed (Figure 2). Peatland plays a positive function as a biogeochemical barrier, which reduces the content of chemical compounds moving in the groundwater throughout the peatland. In groundwater between Ch3 and Ch2, the concentrations of nitrates (38.5%), N-organic (10%), N-total (24.5%), ammonium (38.7%), dissolved total carbon (33.1%), dissolved total inorganic carbon (10%), and dissolved organic carbon (57.5%) were significantly decreased [78].

##### C/N

Another effective and common indicator of peat decomposition is the use of C/N ratios [79]. This indicator is based on the observed residual enrichment of N relative to C during mineralization of organic matter.

The concentrations of N-total in Baltic-type raised bog were significantly lower compared to the N-total in fen and in peat-moorsh soils (Table 1). Baltic-type raised bog receives water and nutrients primarily from atmospheric deposition.

Significant differences of C/N ratios between less decomposed Baltic-type raised bog, fen, and highly decomposed peat-moorsh soils were observed (Table 1). The C/N values in undrained peats were significantly higher as compared to that of corresponding drained peats. The highest C/N ratio and the lowest degree of decomposition in the Baltic-type raised bog was measured.

The high C/N ratios in the deeper layer of peat indicates more intensity in the accumulation of organic matter with the formation of mature HAs as compared to that in the upper layer. In contrast, the low values of the C/N in the surface layer of all peat-moorsh soils correspond to drier conditions and the intensity of organic nitrogen mineralization, leading to the formation of gaseous substances such as ammonia and the further emission of N_2_O and N_2_ into the atmosphere [80,81,82].

Baltic raised bog and fen represent two types peats with similar values of the degree of decomposition H2–H5 (Table 1). Baltic raised bog as nitrogen deficiency system was shown to have lower concentrations of N total and higher C/N ratios compared with that of the corresponding fen. Drainage was shown to be the principal factor causing the increase of the degree of decomposition and the decrease of C/N ratios. Peat-moorsh soils were created as a result of secondary transformation of fen. Therefore, in peat-moorsh soils, higher degrees of decomposition H7–H8, higher C/N ratios, and lower concentrations of TOC compared with that of the corresponding in fen were measured. For all peat-moorsh soils, the values of decomposition degree were shown to be in line with the results calculated for C/N ratios (Table 1).

Our results are in line with the results of Malmer and Holm [83] and of Kuhry and Vitt [84]. They pointed out that lower C/N ratios characterize more decomposed peat material.

According to Broder et al. [85], the lowest humification and high C/N ratios in *Sphagnum* bogs justified the high polyphenol content. On the other hand, Rice and MacCarthy [86] and Anderson [81] pointed out that the C/N ratio does not always reflect soil organic matter mineralization and, moreover, that the percentage share of C_WHE_ would seem to be a better indicator of organic matter mineralization as compared to the C/N ratio (expanding the trend in the C/N ratios with depth), which is bound to hydrological changes of nitrogen concentration.

### 3.2. Analytical Data of HAs

#### 3.2.1. Elemental Composition of HAs

One of the most fundamental characteristics of HAs is its elemental composition expressed in the weight and atomic percentage structure of particular elements [86,87].

The C content ranged from 41.79% to 47.04%, from 40.76% to 44.38%, and from 36.09% to 46.03% in Baltic-type raised bog, fen, and peat-moorsh soils, respectively (Table 6). The C concentration increased with an increase in the depth profile for highly decomposed peat, while the opposite was shown for oxygen content, which decreased in all peat profiles with increasing depth.

The H and N amounts here demonstrated higher variability.

The S concentration decreased with increasing depth in Baltic-type raised bog and fen. In contrast, the concentrations of S increased with increasing depth in peat-moorsh soil compared to the increasing depth in Baltic-type raised bog and fen.

The ratio of H/C, C/N* (in HAs), and O/C bears much more diagnostic information than the elemental composition of HAs. The magnitude of the H/C ratio has been used to indicate the degree of aromaticity (a small value) or aliphaticity (a large value) of a substance [52]. The values of H/C ratio in studied HA samples are in the range from 1.20 to 1.28 in Baltic-type raised bog, from 1.12 to 1.24 in fen, and from 1.00 to 1.25 in peat-moorsh soils (Table 6). However, these H/C and C/N* values among peat soils are not significantly different.

The decrease of the H/C ratio in HAs with increasing depth in undrained peats (Baltic-type raised bog and fen) is related to the peat accumulation rate and development of the aromatic structures, where more labile structures are destroyed or transformed into the appearance of more stable aromatic and polyaromatic structures.

On the other hand, in drained peat (peat-moorsh soil) with increasing depth, the increase of the H/C in HAs increased, indicating less amounts of the C in HAs as compared to that of undrained peats (Baltic-type raised bog and fen).

The C/N* and O/C ratios increased with increasing depth. The high values of C/N* in the deeper layer are related to the presence of proteinaceous materials of living organic matter, indicating a higher intensity of humification than mineralization process with the formation of mature HAs (Table 6).

The significant negative correlations between the O/C vs. C/N* (r = −0.448) and H/C vs C/N* ratio (r = −0.632) indicate that, with an increase in depth, the decarboxylation processes were in line with the increase of the N concentration in HAs (Table 9).

According to Van Krevelen [88], the H/C atomic ratios from 0.7 to 1.5 correspond to aromatic systems coupled with aliphatic chains and contain up to ten carbon atoms. The O/C ratio, for its part, is considered as an indicator of carbohydrate and carboxylic group contents and can be directly related to the aromatization of the peat-forming organic matter. DiDonato et al. [87] suggested that more of the carboxyl-containing aliphatic molecules are sourced from lignin.

#### 3.2.2. VIS spectra of HAs

The E_4_/E_6_ ratio is a valid and informative index for the characterization of aromatic condensation and poly-conjugation in the humic molecule [35,89,90].

Table 7 shows that the E_4_/E_6_ ratio of HAs fraction in peat-moorsh soils ranged from 5.18 to 6.95; in fens, from 4.46 to 6.2; and in Baltic-type raised bog, from 3.98 to 6.36. In peat-moorsh soil profiles, the variability of this value was generally low. As expected, the E_4_/E_6_ ratio of HAs decreased with increasing depth profile in all investigated sites related to progressive humification and condensation of aromatic constituents. These findings are in line with our data of elemental analysis of HAs, indicating more chemically mature HAs in the bottom rather than in the upper layers.

Moreover, the E_4_/E_6_ trend was found to be higher in drained peats compared to E_4_/E_6_ from undrained peats. This reflects a lower degree of aromatic condensation and poly-conjugation and a lower degree of humification in the molecules of HAs from drained rather than undrained peats [88,91,92].

In this context, Kļaviņš and Sire [93] showed strong negative correlations between total acidity values and the E_4_/E_6_ ratios in peat bog profiles. They pointed out that an increase of the acidic groups in the HAs samples resulted in a reduced E_4_/E_6_ ratio.

#### 3.2.3. Electron paramagnetic resonance of HAs

Our study shows the impact of peat type and the degree of decomposition on the EPR spectra, which consists of one signal with ΔH_pp_ at about 4 G typical for radical species (Figure 3) (Table 7).

A strong aromatization in the deeper layer is distinctly observed for the Baltic-type raised bog. The drop of oxidative conditions in the depth has led to an increase in EPR signal intensity for HAs (Table 7) and decline of the g-value from 2.0035 to 2.0021.

The maximum of spin concentrations determined in the deepest layers are in line with the VIS-spectra of HAs and reflects more chemically mature HAs from the Baltic-type raised bog compared with the fen and peat-moorsh soils.

Our results are in line with Czechowski and Jezierski [94], who observed a steady decrease in the g parameter towards the g = 2.0023 value of free electrons for free radicals in bituminous carbon components in parallel with an increase in the aromaticity of the free radicals. In addition, the effect of g parameter reduction (determined by EPR studies) of semiquinone radicals naturally occurring in HAs of increasing aromaticity (Knüpling at al. [95]) was confirmed and explained theoretically by Witwicki and Jezierska [96] on the basis of quantum mechanical calculations of g parameters for model semiquinone radicals.

Inspection of EPR spectra of HAs from fen and peat-moorsh soils (Ch1 and Ch3) reveals g-value characteristics for HAs (2.0036) [64]. The deeper layers of these deposits include an increasing concentration of semiquinone radicals (Table 7) connected with the formation of oxygen containing polymers without growing aromaticity, as the g-values are practically unchanged. In addition, the same g-values of the HAs in all deeper layers are related to similar redox conditions.

The sample Ch2 (peat-moorsh soil) reveals a rather unexpected change of the spin concentration that appeared to be highest in the upper layer. The oxidative conditions of the upper layer impacted the g-value and the spin concentration in this HAs. The g-value is smaller for the upper layer (2.0035) as compared to the g-values (the difference exceeds the experimental error) for lower layers (2.0036), showing a slightly greater aromaticity. This phenomenon is related to the high degree of decomposition of the upper layer; the most transformed peat soil from all studied peat-moorsh soils; the highest O/C ratio in molecule of HAs; and the highest TOC, C_HWE_, and N-total (Table 1). In the context of CH2, due to a high in situ, the oxidative properties in the decomposition rate of the organic matter are faster than its accumulation.

#### 3.2.4. Thermal Properties of HAs

All shapes of DTA and DTG curves are compatible. Thus, each thermal effect on the DTA curve is related to a weight loss measured in TGA of thermal degradation of HAs exhibiting endothermic and exothermic peaks associated with these in all samples (Figure 4; Table 8) [97,98,99]. The DTA curves of all HAs showed one endothermic effect and three exothermic effects.

The endothermic effect ranges from 74 to 84 °C, with weight loss from 12% to 20% and corresponding to the evaporation of absorbed water and dehydration reaction. This effect does not demonstrate any significant changes *downsizing into* the peat profile, and the intensity of weight losses is related to weakly bounded water by hygroscopic components.

Our results are in line with Schnitzer et al. [100], Kodama and Schnitzer [101], Shurygina et al. [102], Leinweber and Schulten [103], and Gołębiowska et al. [65], who pointed out that, at low temperature (about 100 °C), the evaporation of hydroscopic moisture is observed. In addition, Kļaviņš and Sire [59] showed that the amount of hygroscopically bound water in HAs from peat is higher compared to peat from the corresponding layer and does not differ in peat profile.

The thermal analysis of HAs in all peat samples showed an increasing combustion temperature with a downsizing depth of the profile for three exothermic effects (Table 8), which are related to more thermally labile fractions in HAs of the upper layer compared to bottom layers.

The first exothermic reaction ranged from 277 to 320 °C with a mass loss from 27.4% to 31.6% related to the thermal combustion of carbohydrates, peptides, lignin, external functional groups of HAs, decarboxylation of acidic groups in aliphatic compounds, and dehydration of hydroxylate aliphatic structures.

The second exothermic reaction varied from 348 to 427 °C with a mass loss from 9.1% to 32.4%, corresponding to the combustion of less mature components of HAs.

The third exothermic reaction ranged from 437 to 496 °C with a mass loss from 24.6% to 41.4%, which is related to the combustion of bitumens, to highly mature HAs of increasing thermal stability, and to the cleavage of C–C bonds [104,105].

In contrast, in Ch1 and Ch2 (peat-moorsh soils with the highest degree of decomposition), the mass loss of the second exothermic effect increased with an increase in depth.

The lowest mass loss was measured for the second and the highest mass losses for the third exothermic effect in Ch2 with the highest degree of decomposition.

Schnitzer and Levesque. [66] detected two exothermic peaks between 200 and 500 °C for humic substances of podzolic soil. The authors hypothesized that the first peak was associated with dehydratation and decarboxylation and that the second was associated with carbon oxidation.

The thermal analysis provides some important information on HAs structure based on Z parameter values, which expresses the ratio between thermally labile and stable parts in HAs (mainly aliphatic and aromatic) [97].

Inspection of Z parameters revealed that, with increasing depth, HAs contain more structures resistant to oxidation in high temperatures. The HAs from Ch2 and Ch3 (peat-moorsh soils) revealed a more aliphatic character in contrast to the CH1 and Ch4 Baltic-type raised bog and fen, which were found to be more aromatic (Table 8).

The significant negative correlation between the heat of the combustion (DTA_1_/DTG_1_) and E_4_/E_6_ (r = −0.528) (Table 9) may be attributable to more condensed stable aromatic structures of HAs in the deeper layers of peat deposits.

The thermal analysis of HAs in all peat samples showed an increasing combustion temperature with a downsizing depth of the profile for three exothermic effects (Table 8), which are related to more thermally labile fractions in HAs of the upper layer compared to bottom layers.

The first exothermic reaction ranged from 277 to 320 °C with a mass loss from 27.4% to 31.6% related to the thermal combustion of carbohydrates, peptides, lignin, external functional groups of HAs, decarboxylation of acidic groups in aliphatic compounds, and dehydration of hydroxylate aliphatic structures.

The second exothermic reaction varied from 348 to 427 °C with a mass loss from 9.1% to 32.4%, corresponding to the combustion of less mature components of HAs.

The third exothermic reaction ranged from 437 to 496 °C with a mass loss from 24.6% to 41.4%, which is related to the combustion of bitumens, to highly mature HAs of increasing thermal stability, and to the cleavage of C–C bonds [104,105].

In contrast, in Ch1 and Ch2 (peat-moorsh soils with the highest degree of decomposition), the mass loss of the second exothermic effect increased with an increase in depth.

The lowest mass loss was measured for the second and the highest mass loss for the third exothermic effect in Ch2 with the highest degree of decomposition.

Schnitzer et al. [100] detected two exothermic peaks between 200 and 500 °C for humic substances of podzolic soil. The authors hypothesized that the first peak was associated with dehydratation and decarboxylation and that the second was associated with carbon oxidation.

The thermal analysis provides some important information on HAs structure based on Z parameter values, which expresses the ratio between thermally labile and stable parts in HAs (mainly aliphatic and aromatic) [97].

Inspection of Z parameters revealed that, with increasing depth, HAs contain more structures resistant to oxidation in high temperatures. The HAs from Ch2 and Ch3 (peat-moors soils) revealed a more aliphatic character in contrast to the CH1 and Ch4 Baltic-type raised bog and fen, which were found to be more aromatic (Table 8).

The significant negative correlation between the heat of the combustion (DTA_1_/DTG_1_) and E_4_/E_6_ (r = −0.528) (Table 9) may be attributable to more condensed stable aromatic structures of HAs in the deeper layers of peat deposits.

The findings illustrated in Table 9 show a significant negative correlation between TOC concentrations and Z values (r = −0.645), which is related to the increasing contents of aromatic structures in line with the downsizing of peat profile. The above data agrees with E_4_/E_6_ and C/N values (Table 9), where the decrease of both ratios with an increase of depth corresponded to a high degree of aromatic condensation and poly-conjugation of aromatic structures and to the presence of relatively low proportions of aliphatic structures.

As shown in the study of thermal organic matter stability [106], these properties are a function of the chemical composition, degree of humification, and mineral association. However, Kļaviņš and Sire [93] showed that the decay of more labile structure in HAs decreases as the degree of peat decomposition increases, though more condensed aromatic structures were positively correlated with the peat degree of decomposition.

Francioso et al. [107] showed in peat HAs various biochemical fractions of plants preserved during peat formation characteristic of thermal decomposition and more labile structures in the origin and formation process of HAs. This preservation therefore might be the result of anoxic environmental conditions occurring due to peat accumulation.

#### 3.2.5. PCA Analysis

A variety of multiple causes are responsible for the composition of peat and the properties of HAs from undrained and drained peats. Therefore, a comparative study was carried out of the chemical composition of peats and the properties of HAs from various undrained and drained types of peats developed in various compositions of botanical cover, peat-forming species, and oxic and anoxic conditions as a result of the oscillation of ground water during drainage.

The PCA analysis was conducted to determine the impact of various types of peat and decomposition degree on the properties of peat deposits and HAs. The results reveal that the properties of organic matter and HAs vary in different environments (Figure 5).

The results of the PCA explain 80.26% of the total variability of organic matter properties. The first two axes were statistically significant (α < 0.05): the first axis (PC1) explains 45.98%, and the second (PC2) explains 19.84% of the total variability, while the third axis (PC3) explained 14.43% (Table 10). The PC1 was negatively associated with the TOC, C/N, spin concentration DTA_1_/DTG_1_, and C/N* and positively coordinates with E_4_/E_6_, g-value, and carbon in HAs on this axis in peat soils. The PC2 showed a close negative association with C_HWE_ and H/C.

The results of the PCA analysis showed on PC1, with 45.98% variability, a significantly positive correlation between TOC, spin, and DTA_1_/DTG_1_ (Figure 5). The data suggest significantly an increased TOC, spin, and DTA_1_/DTG_1_ with the depth, indicating a decrease in the content of thermolable structural units (carbohydrates, free, and bound functional groups) and an increase in thermostable skeleton part of the HAs molecules. This reflects that the transformation of the organic matter is strongly connected with the humification process and the molecular structure of HAs. Moreover, there were significant negatively associate of TOC, spin, and DTA_1_/DTG_1_ with E_4_/E_6_. The observed changes can be explained by a higher degree of aromatic condensation and poly-conjugation in the molecules of HAs, in the deeper layers (Table 10).

Additionally, on the second axis PC2 with 19.84% variability, the C_HWE_ vs. H/C was significantly positively associated (Figure 4). The decrease of C_HWE_ content and H/C values with the depth suggests the decline of aliphatic bridges between aromatic structural units HAs. On this basis, it can be stated dominantly the accumulation processes rather than ones of decomposition with the depth of peat profile.

The third axis showed a significant positive correlation with ΣDTA/ΣDTG (Table 11). These variables suggest that the first principal component describes the dependence of VIS-spectra, electron paramagnetic resonance, and thermal properties of HAs on the organic matter. Therefore, a higher g-value and E_4_/E_6_ in peat-moorsh soils as well as fens compared to those from Baltic-type raised bogs affects a lower degree of aromatic condensation and humification of HAs.

## 4. Conclusions

This study has examined the vulnerability of organic matter of Baltic-type raised bog, fen, and peat-moorsh soils to decomposition by determining the chemical composition and physicochemical properties of HAs.

Drainage was shown to be the principal factor causing the chemical composition and physicochemical properties of HAs. The latter properties of HAs were found to be in line with their chemical composition of peats.

Conversion of undrained to drained peatlands (modifications in *oxic–anoxic properties)* has led to an increase of the degree of decomposition, relating to the mass loss of organic matter, particularly labile organic fractions, and was confirmed by a correspondingly lower content of TOC, lower C/N values, and higher E_4_/E_6_ values. In line with an increase of the degree of organic matter decomposition, the N-total; E_4_/E_6_, g-values; mass loss for exo1, exo 2, and exo 3; Z values; and spin concentrations all increased while C/N decreased.

In light of these results, the HAs from undrained peatlands were found to be richer in aromatic structures and can be regarded therefore as more humified and mature (therefore, more stable) than that of peat-moorsh soils. Overall, the most important indicators for the propensity of organic matter to decomposition as a consequence of drainage identified in the present study were more aliphatic structures in the HAs. There would appear to be therefore a positive feedback loop of drainage and the consequent vulnerability of soil organic matter to decomposition.

In this context, the analytical methods of thermal analysis together with optical densities and paramagnetic behaviour can be said to be suitable and effective tools for characterizing the origin and formation process of HAs in various environmental properties.

## Figures and Tables

**Figure 1 molecules-25-02587-f001:**
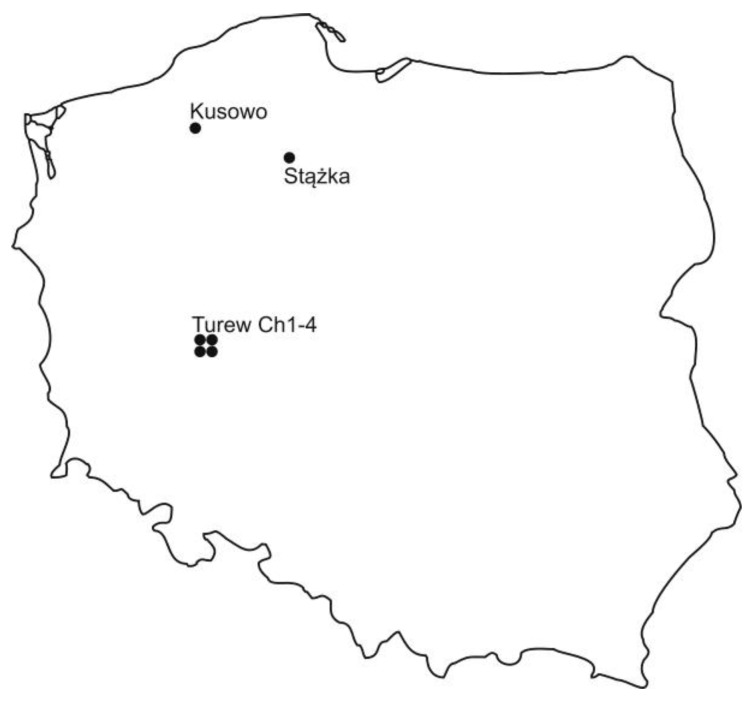
Location of the study sites.

**Figure 2 molecules-25-02587-f002:**
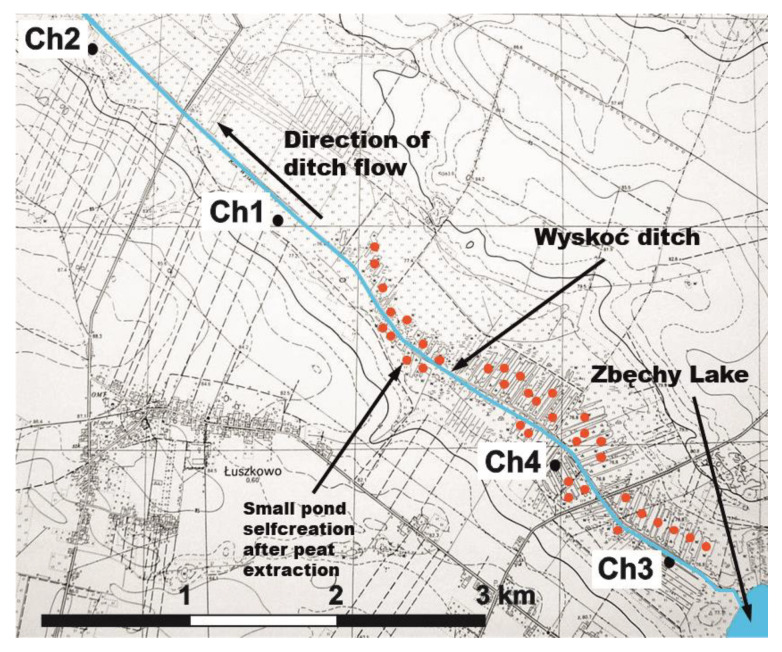
Location of the sites Ch1, Ch2, Ch3, and Ch4 (peat-moorsh soils—Turew).

**Figure 3 molecules-25-02587-f003:**
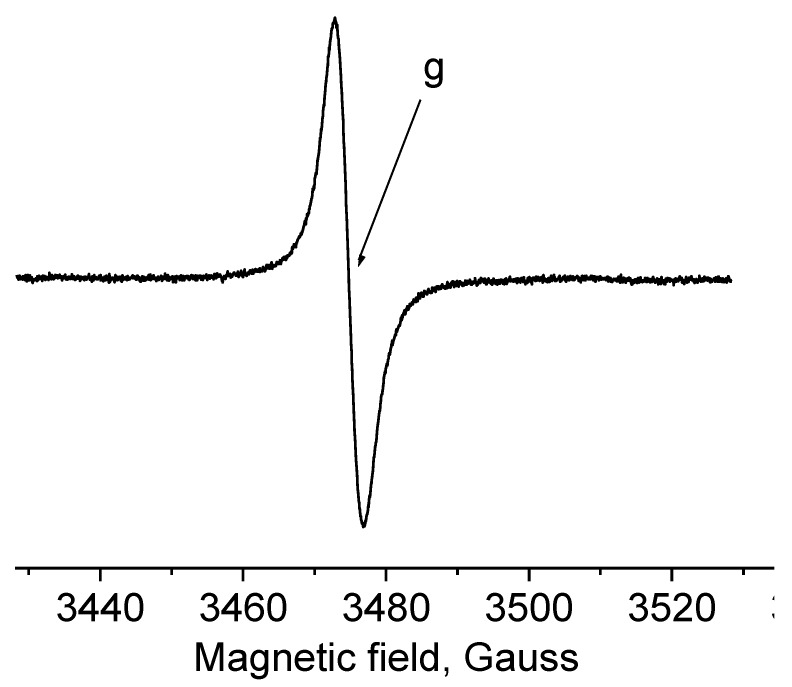
The electron paramagnetic resonance (EPR) spectra of the studied samples of HAs consists of one signal with ΔH_pp_ at about 4 G typical for the radical species.

**Figure 4 molecules-25-02587-f004:**
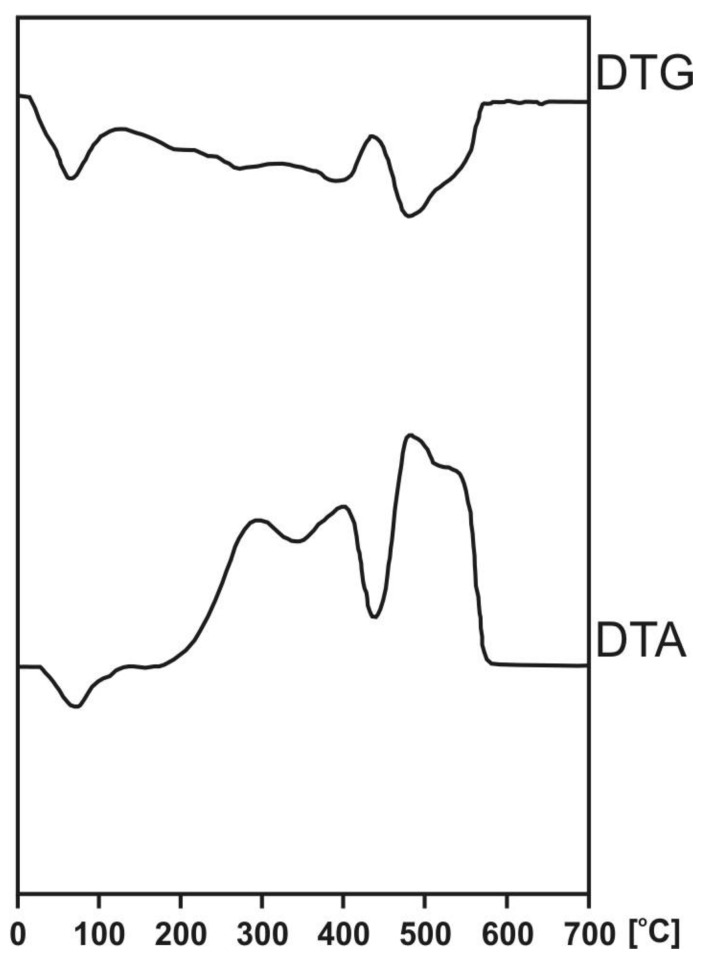
Thermogram of HAs isolated from Kusowo Bog at the depth 25–50 cm.

**Figure 5 molecules-25-02587-f005:**
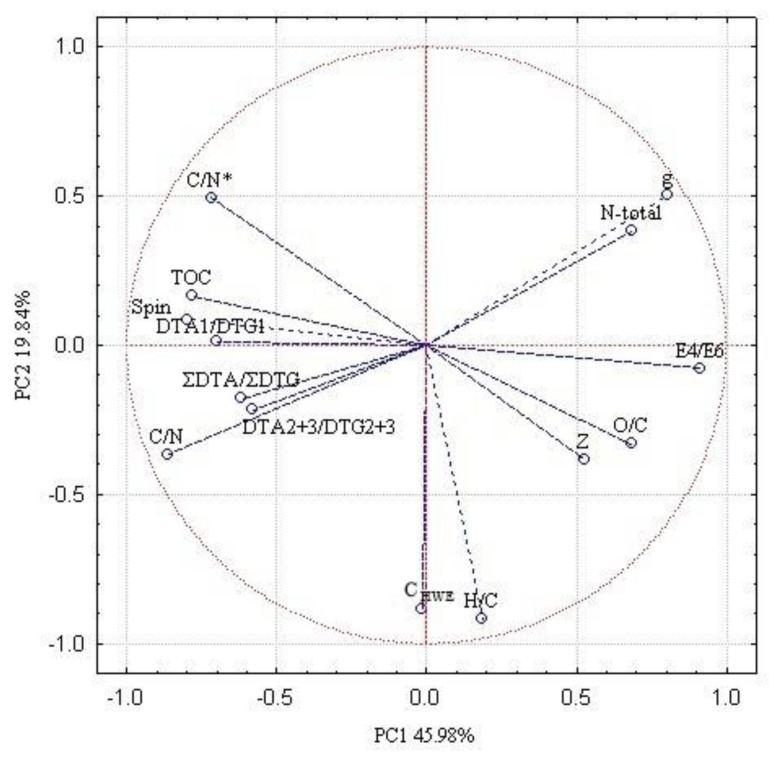
Eigenvectors of soil chemical variables: Abbreviations as in the Table 1, Table 2, Table 3, Table 4, Table 5, Table 6, Table 7, Table 8 and Table 9.

**Table 1 molecules-25-02587-t001:** The classification of peat (WRB 2015), type of peat, degree of decomposition, and chemical properties of peat soils.

Sampling Site		GPS	Classification of Peat (WRB 2015)	Depth (cm)	Type of Peat Based on Macrofossil Analysis	Degree of Decomposition (Von Post)	TOC (g kg^−1^)	C_WHE_ (g kg^−1^)	N-total (g kg^−1^)	C/N
Baltic-type raised bog	Kusowo	53° 48′ 07.83″ N 16° 32′ 42.03″ E	Ombric Hemic Fibric Histosols (Dystric)	0–25	*Sphagnum*	H2	570.05 ± 24.83	15.42 ± 1.59	12.69 ± 1.53	44.92 ± 5.37
				25–50	Cotton grass-*Sphagnum*	H3/H4	575.66 ± 31.34	12.51 ± 1.31	10.83 ± 0.94	53.15 ± 7.30
				50–75	*Sphagnum*, cotton grass-*Sphagnum*	H3	590.44 ± 36.83	12.43 ± 0.38	9.80 ± 0.75	60.25 ± 8.86
				75–100	Cotton grass-*Sphagnum*	H4/H5	596.56 ± 23.60	11.50 ± 1.03	9.42 ± 0.61	63.33 ± 7.51
Fen	Stążka	53° 36′ 17.58″ N 17° 57′ 20.38″ E	Ombric Hemic Fibric Histosols (Dystric Lignic)	0–25	Sedge-*Hypnum*	H3	501.94 ± 37.17	12.49 ± 2.29	20.01 ± 1.26	25.08 ± 4.30
				50–75	Sedge-*Hypnum*	H4	557.51 ± 47.54	5.39 ± 0.72	18.75 ± 2.23	29.73 ± 5.78
				75–100	Sedge	H5	581.50 ± 42.23	5.41 ± 0.47	18.11 ± 2.51	32.11 ± 6.57
Peat-moorsh soils, Turew	Ch1	52° 01′ 35.45″ N 16° 52′ 34.80″ E	Rheic Murshic Sapric Histosols (Limnic Dystric)	0–25	Moorsh soil	H8	294.04 ± 31.96	8.81 ± 0.64	19.19 ± 3.73	15.32 ± 3.19
				25–50	Sedge	H8	445.85 ± 35.80	7.80 ± 0.48	24.38 ± 1.71	18.29 ± 2.51
				50–75	Sedge	H7	498.20 ± 33.03	6.09 ± 0.64	26.66 ± 1.49	18.69 ± 2.16
				75–100	Sedge	H7	496.60 ± 43.24	5.79 ± 0.44	25.62 ± 1.45	19.38 ± 2.25
	Ch2	52° 02′ 21.70″ N 16° 51′ 09.50″ E	Rheic Murshic Sapric Histosols (Lignic)	0–25	Moorsh soil	H8	395.39 ± 35.77	15.89 ± 1.25	31.03 ± 1.65	12.74 ± 2.72
				25–50	Alder swamp	H8	455.81 ± 45.72	10.15 ± 0.89	27.66 ± 2.84	16.48 ± 2.40
				50–75	Sedge	H8	497.46 ± 31.90	7.27 ± 0.70	24.86 ± 2.18	20.01 ± 2.67
				75–100	Sedge	H8	496.11 ± 47.08	5.06 ± 0.53	21.69 ± 2.39	22.87 ± 2.16
	Ch3	52° 00′ 57.50″ N 16° 53′ 49.75″ E	Rheic Sapric Dystric Histosols (Calcic Limnic)	0–25	Moorsh soil	H8	246.63 ± 21.40	7.62 ± 1.45	19.51 ± 2.23	12.64 ± 2.10
				50–75	Alder swamp	H8	466.20 ± 32.60	7.48 ± 0.97	24.50 ± 2.30	19.03 ± 3.55
				75–100	Sedge with wooden	H8	484.90 ± 32.20	5.74 ± 0.68	23.97 ± 1.52	20.23 ± 3.13
	Ch4	52° 01′ 12.61″ N 16° 53′ 23.38″ E	Rheic Sapric Dystric Histosols (Limnic Lignic)	0–25	Moorsh soil	H7	370.90 ± 33.70	11.58 ± 0.91	24.81 ± 3.86	14.95 ± 2.76
				50–75	Sedge	H8	471.21 ± 31.04	8.75 ± 0.82	25.83 ± 2.01	18.24 ± 2.53
				75–100	Sedge	H8	488.46 ± 31.70	7.28 ± 0.69	25.02 ± 1.14	19.52 ± 2.03

TOC—total organic carbon; C_WHE_—hot water extractable organic carbon.

**Table 2 molecules-25-02587-t002:** Annual precipitation from 2009 to 2019 (mm).

Sampling Site	Year											
	2009	2010	2011	2012	2013	2014	2015	2016	2017	2018	2019	Average
Peat-moorsh soils Ch1-4 Turew	576.0	698.1	460.6	534.5	552.3	410.2	485.7	793.3	565.9	425.8	429.9	539.3
Baltic-type raised bog Kusowo	754.2	938.9	673.8	758.4	643.0	661.9	630.5	837.9	1092.2	647.1	723.4	760.1
Fen Stążka	556.0	835.6	487.4	652.9	510.0	472.0	485.4	701.0	885.1	456.0	546.0	598.9

The highest average annual precipitation was measured for Baltic-type raised bog (Kusowo) at 760.1 mm, and the lowest was measured for peat-moorsh soils at 539.3 mm.

**Table 3 molecules-25-02587-t003:** The radiocarbon ^14^C dates for Baltic-type raised bog Kusowo.

Depth [cm]	Age ^14^C Date	Calibrated Range 95.4%	BC/AD
9–10	106.87 ± 0.33 pMC	1694–1919	AD
16–17	111.4 ± 0.36 pMC	1692–1919	AD
33–34	80 ± 30 BP	1690–1926	AD
53–54	40 ± 40 BP	1690–1925	AD
65–66	310 ± 30 BP	1485–1650	AD
76–77	375 ± 30 BP	1446–1633	AD
90–91	550 ± 30 BP	1310–1435	AD
98–99	555 ± 30 BP	1310–1431	AD

**Table 4 molecules-25-02587-t004:** The radiocarbon ^14^C dates for fen Stążka.

Depth (cm)	Age ^14^C Date	Calibrated Range 95.4%	BC/AD
12–13	120.43 ± 0.4 pMC	1689–1928	AD
25–26	195 ± 30 BP	1648–1955	AD
40–41	170 ± 30 BP	1659–1954	AD
54–55	155 ± 30 BP	1666–1953	AD
67–68	1005 ± 30 BP	977–1153	AD
83–84	1125 ± 30 BP	783–991	AD
90–91	1295 ± 70 BP	620–890	AD
105–106	1295 ± 35 BP	655–779	AD

BC—before Christ; AD—anno Domini.

**Table 5 molecules-25-02587-t005:** Moisture content of sampling sites.

Sampling Site		Depth (cm)	Moisture (%)
Baltic-type raised bog (Kusowo)		0–25	91.57 ± 1.92
		25–50	92.31 ± 3.92
		50–75	89.69 ± 1.21
		75–100	91.82 ± 1.44
Fen (Stążka)		0–25	93.92 ± 1.84
		50–75	93.15 ± 3.98
		75–100	93.17 ± 0.86
Peat-moorsh soils (Turew)	Ch1	0–25	66.03 ± 7.55
		25–50	77.62 ± 4.27
		50–75	83.21 ± 2.59
		75–100	83.55 ± 2.60
	Ch2	0–25	74.44 ± 5.88
		25–50	81.18 ± 4.25
		50–75	82.15 ± 1.44
		75–100	79.24 ± 1.73
	Ch3	0–25	61.31 ± 4.49
		50–75	79.80 ± 2.67
		75–100	82.33 ± 1.90
	CH4	0–25	70.29 ± 5.38
		50–75	86.82 ± 1.97
		75–100	86.04 ± 1.47

Drainage significantly impacted the concentration of the moisture in peats. The moisture contents in undrained peats (90–94%) were higher than in drained peats (61–83%) (Table 5).

**Table 6 molecules-25-02587-t006:** Elemental analysis (wt.%) and atomic ratios of humic acids (HAs).

Sampling Site		Depth (cm)	C	H	N	O	S	H/C	C/N*	O/C
**Baltic-type raised bog (Kusowo)**	Kusowo	0–25	43.41	4.66	3.26	47.85	0.82	1.28	15.53	0.83
		25–50	41.79	4.24	2.56	50.78	0.63	1.21	19.04	0.91
		50–75	46.90	4.78	2.65	45.20	0.47	1.21	20.64	0.72
		75–100	47.04	4.74	2.74	45.10	0.38	1.20	20.02	0.72
Fen	Stążka	0–25	40.76	4.24	2.85	51.23	0.92	1.24	16.68	0.94
		50–75	44.01	4.14	2.64	48.55	0.66	1.24	19.44	0.83
		75–100	44.38	3.74	2.08	49.25	0.55	1.12	24.88	0.83
Peat-moorsh soils	Ch1	0–25	42.84	4.50	3.64	48.08	0.94	1.00	13.73	0.84
		25–50	41.62	3.98	2.86	50.66	0.88	1.25	16.97	0.91
		50–75	46.03	4.31	2.69	45.86	1.11	1.14	19.96	0.75
		75–100	44.63	4.27	2.86	46.90	1.34	1.12	18.20	0.79
	Ch2	0–25	36.09	3.73	2.86	56.36	0.96	1.14	14.72	1.17
		25–50	42.66	4.13	3.17	48.90	1.14	1.23	15.69	0.86
		50–75	42.61	4.07	2.93	49.00	1.39	1.15	16.96	0.86
		75–100	43.19	4.08	2.90	48.15	1.68	1.14	17.37	0.84
	Ch3	0–25	41.43	4.27	3.45	49.60	1.25	1.13	14.00	0.90
		50–75	42.13	4.10	3.03	48.80	1.94	1.23	16.21	0.87
		75–100	43.41	4.03	2.73	47.58	2.25	1.24	18.54	0.82
	Ch4	0–25	40.57	4.07	3.00	51.15	1.21	1.16	15.77	0.95
		50–75	43.21	4.11	3.07	48.21	1.40	1.11	16.41	0.84
		75–100	43.21	4.03	3.11	48.00	1.65	1.19	16.20	0.83

C/N* ratio in HAs.

**Table 7 molecules-25-02587-t007:** Parameters of VIS-spectroscopy and electron paramagnetic resonance of HAs.

Sampling Site		Depth (cm)	E_4_/E_6_	g-Value	*Spin Concentration × 10^17^
BalticRaised bog	Kusowo	0–25	6.36	2.0035	1.29
		25–50	4.88	2.0029	3.13
		50–75	4.00	2.0021	3.92
		75–100	3.98	2.0023	5.64
Fen	Stążka	0–25	5.67	2.0035	1.04
		50–75	5.57	2.0036	1.38
		75–100	4.46	2.0036	3.41
Peat-moorsh soils	Ch1	0–25	6.78	2.0036	0.77
		25–50	6.12	2.0036	1.71
		50–75	5.94	2.0036	2.22
		75–100	6.19	2.0036	2.77
	Ch2	0–25	6.95	2.0035	2.20
		25–50	6.73	2.0036	1.23
		50–75	6.49	2.0036	1.43
		75–100	5.59	2.0036	1.12
	Ch3	0–25	6.65	2.0036	0.95
		50–75	6.14	2.0036	2.27
		75–100	5.18	2.0036	3.82
	Ch4	0–25	6.37	2.0035	1.48
		50–75	6.19	2.0035	3.34
		75–100	5.70	2.0035	4.30

g-value, the spin concentrations are given in spins/g units, e.g., the last value is 3.41 × 1017 spins/g organic matter.

**Table 8 molecules-25-02587-t008:** Parameters of thermal decomposition of HAs.

Sampling Site		Depth (cm)	Temperature of Effects on DTA Curves (°C)				Mass Loss Corresponding with the Effects on DTG Curves (%)				The Ratios of Area under DTA and DTG Curves for Exothermal Effects			Z $\frac{\mathrm{endo}+\mathrm{exo}1}{\mathrm{exo}2+3}$
			endo	exo 1	exo 2	exo 3	Endo	exo 1	exo 2	exo 3	$\frac{{\mathrm{DTA}}_{1}}{{\mathrm{DTG}}_{1}}$	$\frac{{\mathrm{DTA}}_{2+3}}{{\mathrm{DTG}}_{2+3}}$	$\frac{\sum \mathrm{DTA}}{\sum \mathrm{DTG}}$	
Raised bog	Kusowo	0–25	74	297	403	489	13.6	27.4	24.0	35.0	3.32	4.08	3.84	0.70
		25–50	75	294	427	485	15.4	28.8	24.1	31.7	3.41	6.10	5.18	0.79
		50–75	76	295	400	473	13.2	29.4	24.6	32.8	4.07	7.83	6.55	0.74
		75–100	74	294	418	490	15.7	28.8	24.7	30.8	4.75	8.02	6.90	0.80
Fen	Stążka	0–25	76	294	390	476	16.0	28.1	22.2	33.7	2.60	2.92	2.81	0.79
		50–75	74	294	399	490	15.7	28.8	24.7	30.8	2.48	3.92	3.29	0.67
		75–100	77	304	419	486	13.5	28.1	28.5	29.9	2.53	3.38	3.44	0.71
Peat-moorsh soils	Ch1	0–25	79	287	365	467	12.7	31.6	25.4	30.4	3.07	4.82	3.88	0.79
		25–50	79	290	372	461	12.7	29.5	28.4	29.4	3.80	5.46	4.50	0.76
		50–75	78	320	421	483	12.6	27.8	32.2	26.3	4.62	6.75	5.31	0.69
		75–100	76	314	426	496	14.2	28.8	32.4	24.6	2.76	5.65	4.68	0.76
	Ch2	0–25	76	277	348	437	21.7	29.1	9.1	40.1	2.15	5.99	4.64	1.03
		25–50	75	281	350	439	17.1	30.0	10.9	41.4	2.99	6.78	5.38	0.92
		50–75	75	294	370	450	18.3	28.0	14.4	39.3	2.24	6.06	4.76	0.86
		75–100	77	292	367	450	18.1	28.8	16.0	37.1	3.08	7.47	5.94	0.88
	Ch3	0–25	80	291	377	476	17.3	30.2	19.8	33.1	2.50	3.58	4.59	1.08
		50–75	84	296	394	481	13.9	28.3	21.3	36.6	3.23	3.70	4.87	0.95
		75–100	81	293	391	480	15.9	31.2	19.3	32.1	4.58	5.17	6.23	0.91
	Ch4	0–25	79	289	367	458	16.4	30.7	16.2	36.7	2.32	4.41	3.64	0.89
		50–75	78	302	394	479	13.7	29.9	23.2	33.2	2.60	4.03	3.38	0.76
		75–100	79	300	399	482	12.1	30.4	28.1	29.4	4.22	7.57	6.44	0.71

DTA—differential thermal analysis; DTG—differential thermogravimetry; Z—the loss ratio of HAs in the low range to those in the high temperature range; endo—endotermic; exo—exotermic effect.

**Table 9 molecules-25-02587-t009:** Correlation coefficients between chemical properties of peat deposits, VIS-spectroscopy, electron paramagnetic resonance, and thermal decomposition of HAs.

			Peat Deposit				HAs						
Parameter	TOC	C_WHE_	N-Total	C/N	E_4_/E_6_	g-Value	Spin Concentr.	$\frac{{\mathrm{DTA}}_{1}}{{\mathrm{DTG}}_{1}}$	$\frac{{\mathrm{DTA}}_{2+3}}{{\mathrm{DTG}}_{2+3}}$	$\frac{\sum \mathrm{DTA}}{\sum \mathrm{DTG}}$	Z	H/C	C/N*
C_WHE_	0.065	-											
N-total	−0.481 *	−0.301	-										
C/N	0.736 *	0.384	−0.900 *	-									
E_4_/E_6_	−0.726 *	0.039	0.697 *	−0.806 *	-								
g-value	−0.480 *	−0.420	0.729 *	−0.865 *	0.744 *	-							
Spin conc.	0.552 *	−0.059	−0.313	0.548 *	−0.756 *	−0.620 *	-						
$\frac{{\mathrm{DTA}}_{1}}{{\mathrm{DTG}}_{1}}$	0.358	−0.103	−0.305	0.426	−0.528 *	−0.454 *	0.612 *	-					
$\frac{{\mathrm{DTA}}_{2+3}}{{\mathrm{DTG}}_{2+3}}$	0.283	0.134	−0.106	0.374	−0.313	−0.584 *	0.552 *	0.607 *	-				
$\frac{\sum \mathrm{DTA}}{\sum \mathrm{DTG}}$	0.228	−0.008	−0.179	0.376	−0.425	−0.569 *	0.669 *	0.759 *	0.850 *	-			
Z	−0.645 *	0.186	0.347	−0.406	0.429	0.159	−0.239	−0.313	−0.126	0.104	-		
H/C	−0.303	0.772 *	−0.326	0.203	0.266	−0.285	−0.339	−0.095	−0.005	−0.051	0.295	-	
C/N *	0.743 *	−0.322	−0.382	0.530 *	−0.815 *	−0.369	0.553 *	0.296	0.152	0.163	−0.532 *	−0.632 *	-
O/C	−0.454 *	0.432	0.458 *	−0.462 *	0.536 *	0.379	−0.421	−0.590 *	−0.307	−0.384	0.570 *	0.279	−0.488 *

* significant correlation coefficient α = 0.05; abbreviations as in Table 1, Table 2, Table 3, Table 4, Table 5, Table 6, Table 7 and Table 8.

**Table 10 molecules-25-02587-t010:** Analysis of principal components for Baltic-type raised bog, fen, and peat-moorsh soils.

Principal Components	Eigenvalues	% of total Variance	Cumulative Eigenvalues	Cumulative % of Variance
PC1	6.44	45.98	6.44	45.98
PC2	2.78	19.84	9.22	65.83
PC3	2.02	14.43	11.24	80.26

**Table 11 molecules-25-02587-t011:** Factor loadings and explained variance of three principal components in PCA.

Variable	PC1	PC2	PC3
TOC	−0.78	0.17	−0.33
C_WHE_	−0.01	−0.89	−0.20
N-total	0.68	0.38	0.46
C/N	−0.86	−0.37	−0.32
E_4_/E_6_	0.92	−0.08	0.18
g-value	0.80	0.50	0.01
Spin concentration	−0.80	0.09	0.30
$\frac{{\mathrm{DTA}}_{1}}{{\mathrm{DTG}}_{1}}$	−0.70	0.01	0.47
$\frac{{\mathrm{DTA}}_{2+3}}{{\mathrm{DTG}}_{2+3}}$	−0.58	−0.22	0.63
$\frac{\sum \mathrm{DTA}}{\sum \mathrm{DTG}}$	−0.62	−0.18	0.74
Z	0.52	−0.38	0.41
H/C	0.19	−0.92	−0.12
C/N*	−0.72	0.49	−0.27
O/C	0.68	−0.33	−0.03

C/N*—atomic ratios of the HAs; abbreviations as in Table 1, Table 2, Table 3, Table 4, Table 5, Table 6, Table 7, Table 8 and Table 9.

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
