# Peer review of "Influence of Drainage on Peat Organic Matter: Implications for Development, Stability, and Transformation"

_molecules, 2020, doi:10.3390/molecules25112587_

Round 1

Reviewer 1 Report

This paper represents results of investigation of the molecular structures of the humic acids, extracted from the raised bog, fen and peat-moorsh soils developed in various conditions. The selected objects are quite interesting in the context of the reclamation of peat bogs and the potential degradation of organic matter.

Nevertheless, few comments are made, which will have to be taken into account before recommendation for further publication:

P.1–L 25,29 and P.18–L58: There is no data in the text about O,N-alkyl

P.1, L 28: It is not clear where the authors came to the conclusion about carboxyl groups? If from the data of elemental analysis, then, in addition to carboxyl groups, it can also be hydroxyl groups of alcohols and carbohydrates ...

Last sentence of section 2.2.: The text does not include soil classification according to WRB. Need to add

Section 2.4: It is necessary to add a short description of the extraction of humic acids, including their purification and drying conditions.

Section 2.5: Please specify how hygroscopic humidity and ash content of humic acids were taken into account?

Table 2. need to moved in the "Results" section

Table 2: H/C, C/N*, O/C - are these likely a molar rations? Need to clarify

Tables 2, 5, 7 and Section 3.1.2.: “C/N* ratio in HA.” H/C, O/C - are these rations in peat samples?

Section 3.1.: “Characteristics of peats”, but Sections 3.1.2.-3.1.5 – Analytical date of HAs

Section 3.2. missing

Author Response

Dear Mrs. or Mr. Reviewer.

Thank you for your help, comments, suggestions, and advices. They were pertinent.

In the attached "Review Report Form" I provide only brief answers to your questions.

Full answers were included in the text of the manuscript and yellow marked.

In the revised version of the manuscript I have added many new analytical methods, tables, figures and explanations as well as references.

Together, they all better explain the processes and mechanisms in peat during drainage.

"Please see the attachments”.

You will find two attachments:

1/ "Review Report Form".

2/ Our new version of the manuscript, which contains changes. All changes are marked in yellow.

Sincerely yours

Lech Wojciech Szajdak

Reviewer 2 Report

Review of the paper entitled: “Influence of drainage on peat organic matter: implications for development, stability and transformation”, written by Szajdak et al., submitted to Molecules.

The goal of this paper is to investigate the evolution of organic matter composition in various peatland stages under different botanical cover, taking into account the soil oxic/anoxic conditions resulting from groundwater oscillations.

The introduction:

The authors may also underline the importance the peatlands for climate change, as peatland are the largest natural terrestrial carbon store. Moreover, damaged peatlands may contribute about 10% of greenhouse gas emissions.

It is also important to note that draining peatland reduces the quality of drinking water due to pollution from dissolved compounds, and may also result in biodiversity loss.

I was wondering if the term of melioration is the most appropriate. Would the authors mean improvement, restoration, preservation?

The goals of this study should be more precise: the authors investigated the evolution of organic carbon composition in 6 different peat soils, located in Poland. This study investigates specific peat soils. At the first lecture of the introduction, we would think about a review giving a toolboc for interpreting organic carbon results. However, I was a little disappointed not to see any more extensive comparisons with the literature.

In general, the results are not enough described, as for decomposition degree, TOC, C/N… Nearly no values are given. The authors may add some values for concentration… The discussion for peat characteristics is relatively tiny and decipher to be extended. It would have been interesting to have humic acids contents for each sample, not only their chemical composition in order to appreciate either an increase or a decrease of their content.

I would suggest to separate the Results and Discussion sections in order to better characterize the peat degradation based on chemical/spectroscopic (E4/E6)/ HA results.

The conclusion is a little speculative on the content of alkyl- and carboxyl- carbon. This has been introduced in the 4th paragraph of the conclusion and in the abstract, but not discussed earlier. This deserves to be introduced, discussed and properly justified in the discussion. Do the authors have 13C NMR data?

At this stage, I would suggest major revision before publication. I suggest some (large) modifications in the structure of the paper (separation of Results and Discussion) and a larger review of literature for discussion of the results. In the following part of the review are some suggestions and questions.

Specific comments

  • 47: Two major types of peatlands.
  • 47: the authors may also add that bogs have an ombrotrophic vegetation.
  • 49: slows decomposition.
  • 51-52: here the authors suggest a third type of peatlands (swamps): they should directly list the 3 types. This sentence should be included in the previous paragraph (L. 47-50).
  • As the authors talk about the agricultural use of peatlands, they also may list other use, and highlight the global percentage of agricultural use. As example, 14% of European peatlands are used for agriculture.
  • 53-59: in this paragraph, the authors may be more precise on the induced processes: the drainage of peatlands induces the mineralization and anthropogenic mixing with mineral soils, a secondary transformation of the peat… (Saürich et al., 2019).
  • 60-63: the sentence may be moved into the previous paragraph (L.59).
  • 66: Ref. 24 - be careful with Polish reference, as the audience may not read Polish.
  • 67-68: please replace inches by centimeters, as we use international unit system.
  • 67-70: the authors underline the accumulation rates of organic matter in soils. They may also add the accumulation rates of restored peatlands, which are higher than drained ones.

Methods

- L. 81-82: Do the authors have an idea of the peatland age? A sample map may also be welcome.

2.2. Collection of samples (no more line numbers).

- Two paragraphs about raised bogs have to be merged. Do the authors suggest that a 657 mm of precipitation is a high rainfall rate?

- What is the pH of groundwater (~4-5?)?

2.3. OC, N, C/N

- can the authors be more precise on how they measure TOC in bulk peat samples: mix, grind, sample size…

- how long did it take for the samples to air dry? What was the soil humidity?

2.4. Extraction and purification of HA

- the method used may be summarized in this part.

2.8. Differential thermal analysis of Has

Instead of TG, the authors should write TGA (thermogravimetry analysis) as written in the paper (section 3.1.5).

2.9. Statistical analysis

- did the authors only use 2 principal components? Are they more than 2 PC with an eigenvalue>1? As at least 3 PC were used in this paper, this should be rewritten in the Methods part.

Results and Discussion

Decomposition degree

  • The results of the decomposition are not really described.
  • “Our knowledge of the degree of decomposition will provide soil chemists chemical information about the physicochemical properties and features of peat soils.” I would change this sentence, as chemical characteristics of peat soil may be translated into peat physico-chemical properties.
  • It is important to more describe the results for the different peat soils, as the raised bog presents the decomposition degree from H2 to H5, indicating an almost entirely undecomposed peat to a moderately decomposed peat over 1 m. Moreover, this peat thickness may reach 12 m, which is largely more than the other samples. All these data have to be carefully compared relatively to the peat thickness and water table level or hydraulic conductivity. The peat age may also explain the difference between all these samples. Do the authors have this information?

TOC

  • “The lower TOC content in the upper layer may be related to the increased rate of decomposition rather than accumulation”: the compaction and the molecular composition may also explain a TOC content evolution with depth.

CHWE

The use and significance of CHWE should be explained in the Methods section. The authors should precise that hot water extractable organic carbon represents the most labile organic carbon of a peat soil, and thus become a significant source of carbon fueling microbial metabolism.

The results should be presented by giving some values. What does the labile (CHWE) fraction represent compared to the total fraction? It represents between 2 and 3% of TOC. Peat moorsh soils have the half of CHWE relatively to raised bog soils, for example. Then, the authors can present the evolution (increase/decrease).

C/N

It is really important to give some numerical values, not only relative values… Undrained peats have nearly 3 times higher C/N than peat moorsh soils (63 vs. 20).

It would also be interesting to compare C/N ration of peat with bog/fen vegetation.

The authors may also explain the higher C/N values of bogs with a lower N content than fen, because bogs are nitrogen-deficient ecosystems

In lot of peat, the C/N ration tend to decrease with depth (example: in the USA), while this study shows the contrary. Some other studies display the same results (in tropical peats) These results may be more discussed.

  • “The high C/N ratios in the deeper layer of peat indicates more intensity in the accumulation of

organic matter with the formation of mature HA as compared to that in the upper layer.” As neither HA content nor the chemical composition of HA have been presented do far, it Is too early to speak of the HA maturity. This can be approached in the following paragraph.

Table 2

  • Why did the authors use C/N ratio for HA, while in most papers, the N/C ratio is used. This would facilitate the comparison with other studies.

3.1.5. Thermal properties of HA

- what does this sentence mean? : “All shapes of DTA and DTG curves are compatible”.

Figure 3

  • DOC concentration is represented on the figure, while it was not discussed in the text. Is it DOC in groundwater? Or does DOC represent CHWE?

3.1.6. PCA analysis.

In addition to eigenvalues of variables, the samples should also be added: this may help to distinguish the control of environmental parameters. In this case, bog, fen and moorsh should be isolated.

Conclusions

The fourth paragraph speculates on the composition of organic carbon (“In the light of these results, the HAs from undrained peatlands were found to be richer in aromatic, alkyl and carboxyl carbon atoms, but contained lower amounts of hetero-alkyl C, exhibiting in evidence a higher degree of aromatic condensation and poly-conjugation than drained peatlands”). This has not suggested in the discussion and should be introduced earlier, not only in conclusion, neither in the abstract.

Author Response

(The authors gave the same response as above.)

Round 2

Reviewer 1 Report

The authors corrected all the comments. The article can be published.